# Real-World Outcomes of Systemic Therapy in Japanese Patients with Cancer (Tokushukai REAl-World Data Project: TREAD): Study Protocol for a Nationwide Cohort Study

**DOI:** 10.3390/healthcare10112146

**Published:** 2022-10-28

**Authors:** Rai Shimoyama, Yoshinori Imamura, Kiyoaki Uryu, Takahiro Mase, Yoshiaki Fujimura, Maki Hayashi, Megu Ohtaki, Keiko Ohtani, Nobuaki Shinozaki, Hironobu Minami

**Affiliations:** 1Department of General Surgery, Shonankamakura General Hospital, Kamakura 247-8533, Japan; 2Department of Medical Oncology and Hematology, Kobe University Graduate School of Medicine, Kobe 650-0017, Japan; 3Department of Medical Oncology, Yao Tokushukai General Hospital, Yao 581-0011, Japan; 4Department of Breast Surgery, Ogaki Tokushukai Hospital, Ogaki 503-0015, Japan; 5Tokushukai Information System Inc., Osaka 530-0001, Japan; 6Mirai Iryo Research Center Inc., Tokyo 102-0083, Japan; 7deCult Co., Ltd., Hatsukaichi 739-0413, Japan; 8General Incorporated Association Tokushukai, Tokyo 102-0074, Japan; 9Cancer Center, Kobe University Hospital, Kobe 650-0017, Japan

**Keywords:** real-world data, cancer chemotherapy, National Cancer Registry, diagnosis procedure combination data

## Abstract

Cohort studies using large-scale databases have become increasingly important in recent years. The Tokushukai Medical Group is a leading medical group in Japan that includes 71 general hospitals nationwide from Hokkaido to Okinawa, with a total of 18,000 beds, and a unified electronic medical record system. This retrospective cohort study aims to evaluate the real-world outcomes of systemic therapy for Japanese patients with cancer using this merit of scale. All adult patients with cancer who received systemic therapy using a centrally registered chemotherapy protocol system at 46 hospitals from April 2010 to March 2020 will be identified (~48,850 patients). Key exclusion criteria include active double cancer and inadequate data extraction. Data will be obtained through electronic medical records, diagnosis procedure combination data, medical prescription data, and the national cancer registration system that includes sociodemographic variables, diagnostic and laboratory tests, concomitant drug prescriptions, cost, and overall survival. Kaplan–Meier estimates will be calculated for time-to-event analyses. Stratified/conventional Cox proportional hazards regression analyses will be conducted to examine the relationships between overall survival and related factors. Our findings provide important insights for future research directions, policy initiatives, medical guidelines, and clinical decision-making.

## 1. Introduction

The first published randomized controlled trial (RCT) was conducted in 1948 to find the standard clinical treatment with streptomycin for tuberculosis [1]. Since then, RCTs have been regarded as the gold standard for addressing various clinical questions, including cancer treatment [2,3]. RCTs generally include highly selected patients with good performance status but exclude patients with complications, such as those with cardiac, renal, and hepatic diseases, and the elderly [4,5]. These conditions lead to discrepancies between the actual clinical practice and RCT outcomes. First, data on clinical outcomes and adverse events in elderly patients and patients with complications are difficult to obtain [6,7]; second, owing to low realizability, the traditional RCT design may be less suitable for emerging technologies, such as those that focus on rare genetic mutations [8]; and finally because survival data from RCTs are generally based on a very narrow observation period, and the extrapolation of the survival curve may fail to allow for the prediction of the long-term treatment effect [9].

In recent years, cohort studies using large-scale databases have become widespread. Real-world data (RWD) are defined as data related to the patients’ health status and treatment history that are collected during daily medical practice, such as data derived from electronic medical records and from hospitalization data [10]. The potential advantages of RWD are demonstrated by the limitations of the ongoing reliance on RCTs [11,12,13]. Consequently, RWD are used in various situations, such as the clinical development of drugs and post-marketing safety evaluation. The Food and Drug Administration in the United States uses RWD to expand the indications for approved drugs [14]. To facilitate the development of a rich, objective, and less biased dataset integrating a broad spectrum of evidence, protocol papers using RWD have been prepared over the past few years [15,16].

The Tokushukai Medical Group is a leading medical group in Japan that includes 71 general hospitals nationwide from Hokkaido to Okinawa, with a total of 18,000 beds, accounting for approximately 1.2% of all hospital beds in Japan [17]. The electronic medical record system that we use is unified, and medical coding of medication names and laboratory test values is standardized. This medical record system is managed by Tokushukai Information System Inc., and the patients’ medical information can be extracted from each hospital’s patient registration system. For systemic cancer therapy, a shared chemotherapy protocol system was installed in 51 hospitals by the end of March 2020. All prescribed chemotherapy regimens are approved by the regimen review committee of the Tokushukai Medical Group; therefore, the treatment schedule and dosage are unified nationwide. Among these hospitals, 46 hospitals applied the diagnosis procedure combination data (DPC), which is the Japanese claims record for acutely hospitalized patients, including information on the main diagnosis, interventions, and comorbidities. The DPC was developed as a measurement tool aimed at making acute hospital care transparent, standardizing Japanese medical care, and evaluating and improving its quality [18].

The Tokushukai REAl-world Data (TREAD) project is being developed as an observational population-based retrospective cohort study that aims to collect RWD on systemic therapy for Japanese patients with cancer in the Tokushukai Medical Group with this merit of scale.

Here, we report the main protocol of the study that will generate data through several sub-projects for each type of cancer to assess whether the results of clinical trials extend to the real world. Moreover, the data will enable the examination of temporal trends in treatment patterns, as well as treatment outcomes and prognostic factors that affect overall survival. 

## 2. Materials and Methods

### 2.1. Study Population

This study was approved by the Ethics Committee of Tokushukai Medical Group in April 2020 (no. TGE01427-024). All adult patients with cancer who received centrally registered systemic therapy at 46 DPC hospitals within the Tokushukai Medical Group from 1 April 2010 to 31 March 2020 will be identified within each of the sub-projects containing an analysis of various cancers involved in this study. The inclusion and exclusion criteria are presented in Table 1.

### 2.2. Information Source and Data Collection

Information on the patients who met the inclusion criteria will be extracted by the Tokushukai Information System Inc., while researchers will be blinded to statistical analyses. Figure 1 provides an overview of how variables will be collected.

Patient demographic data, including age, birth year and month, sex, postal code, and insurance type, were documented at the time of the issuance of medical record identification. In addition, body height, body weight, body mass index, body surface area, and date of measurement were recorded based on practical requirements. Moreover, the date of the last visit, the last date of survival confirmation, the date of death, and the diagnosis on medical receipt were automatically recorded as part of routine clinical practice. The above information can be extracted from a unified medical record system (e-Karte and Newtons2, Software Service, Inc., Osaka, Japan).

Treatment information, including inpatient and outpatient therapies, particularly chemotherapy regimens, can be retrieved from a shared chemotherapy protocol system (SSI srvApmDrop, Software Service, Inc., Tokyo, Japan). This system links the start and end date, the date of administration, the treatment interval, the number of treatment cycles, the dose of administration, the data of dose adjustment, and the performance status.

Data from the Japanese National Cancer Registry [19,20] are available for all hospitals, including the diagnosis (date, site, laterality, pathology, and differentiation), staging (clinical and pathological TNM classification [21]), date of the first visit, treatment details (surgery, endoscopic treatment, radiotherapy, chemotherapy, and endocrine therapy), and survival outcomes (last date of survival confirmation, date of death, and cause of death).

Patient, treatment, and cancer registry data are combined with anonymized identification and cleaned by Tokushukai Information System Inc. (Osaka, Japan). Patients with double cancer and/or diagnostic errors will be excluded from the study. Treatment history will be organized, and missing data will be supplemented by a direct check of medical records in cases of inconsistency or lack of information. Patients with unknown treatment history will be also excluded.

Details of hospitalization information such as diagnosis at admission (ICD-10), type of admission (scheduled or emergency), date of admission and discharge, outcome at discharge, complication (ICD-10), costs, and history of smoking (Brinkman index) can be extracted from the DPC data, which serves as the National Hospital Discharge Database for clinical studies in Japan [18,22].

Prescription data, such as the name and dose of the medicine, as well as the date and days of prescription, can be added from the electronic ordering system (e-Karte and Newtons2). The following treatment agents will be assessed in this study: (1) antitumor agents, (2) antihypertensive agents, (3) antithrombotic and antiplatelet agents, (4) antidiabetic agents, (5) antilipidemic agents, (6) antibacterial/antiviral/antifungal agents, (7) antiacid agents, (8) steroids, and (9) probiotics.

Furthermore, laboratory test information, such as blood cell count, biochemical tests, tumor markers, coagulation/fibrinogenic tests, infectious diseases, and urinalysis, can be retrieved from the electronic ordering system (e-Karte and Newtons2). The detailed parameters are listed in Table 2. Admission, prescription, and laboratory test data are combined with basic patient information with anonymized identification and cleaned again by Tokushukai Information System Inc. The final anonymized datasets, created as described above, will be handed over to statistical analysts with simplified case identification.

The analysis will be performed using statistical software (R, SAS, SPSS, or other statistical software) on computers of internal and external research institutes (Kobe University (Kobe, Japan) and deCult Co., Ltd. (Hatsukaichi, Japan)). Data will not be shared, except among computers that process and analyze them.

### 2.3. Study Endpoints

For this study, the primary endpoint is overall survival, whereas the secondary endpoints are the time to treatment failure, recurrence-free survival, and adverse events, evaluated based on the Common Terminology Criteria for Adverse Events v.5.0 Japanese translation—Japan Clinical Oncology Group (JCOG) version (CTCAE v. 5.0-JCOG) [23].

### 2.4. Sample Size Calculation

The sample size for each cancer type is estimated based on the actual number of patients rather than specific hypotheses. Only 27 hospitals were equipped when the chemotherapy protocol system was started in April 2010, and the number of hospitals where this protocol system was used increased to 51 in March 2020. During this study period, the number of registered regimens increased rapidly, reflecting advances in oncology. These dynamic shifts preclude simple sample-size calculations. Alternatively, each sample size over the study period can be estimated based on the number of patients receiving drugs through the protocol system in 2019 (176 for biliary tract cancer, 1534 for breast cancer, 2324 for colorectal cancer, 212 for esophageal cancer, 956 for gastric cancer, 72 for kidney cancer, 1473 for lung cancer, 319 for ovarian cancer, 1026 for prostate cancer, 501 for pancreatic cancer, 71 for uterine body cancer, 83 for uterine cervix cancer, 199 for urinary tract cancer, 104 for gastrointestinal stromal tumors, and 123 for liver cancer). The expected sample sizes are listed in Table 3.

### 2.5. Statistical Analysis

Basic statistics (absolute and relative frequencies for categorical variables; quartiles, maximum, minimum, and mean for continuous variables; and quartiles and relative frequencies for discrete variables) will be obtained to summarize the distribution of variables related to patient background factors, complications, other prognostic factors, and endpoints.

Kaplan–Meier curves will be obtained for each stratum consisting of patient background and prognostic factors for the occurrence of events associated with various endpoints (overall survival, time to treatment failure, recurrence-free survival, and adverse events). Several univariate logistic regression analyses will be performed, with the presence or absence of events in each stratum as the objective variable and the categorical value of each background or prognostic factor as the explanatory variable.

A hierarchical set of predictive models will be constructed by combining explanatory variables that are expected to contribute a single-layer or multi-layer proportional hazards model, which will be set up by incorporating each predictive model, and Cox multiple regression analysis will be performed. If proportional hazards are not acceptable, parametric hazard models such as the Cure–Weibull gamma-frailty model or the mixed Weibull gamma-frailty model will be applied for survival time analysis (including in combination with logistic regression models) [24]. Based on the likelihood (or partial likelihood in the case of Cox regression), the Akaike information criterion (AIC) and Bayesian information criterion (BIC) will be used to explore the optimal model (if the number of cases covered differs between models, the average AIC and BIC per case are used instead).

Regarding the estimated value of the regression coefficient of the explanatory variable included in the optimal model, if *p* value is <0.05, it will be judged as an important prognostic factor, and the 95% confidence interval of the hazard ratio regarding the magnitude of the contribution of that factor will be calculated.

### 2.6. Patient and Public Involvement Statement

This study is a retrospective study based on past medical records, and there is no involvement of patients or the general public.

## 3. Discussion

We established a method for extracting real-world outcomes of systemic cancer therapy and its background factors using the Tokushukai medical database. Following the main protocol, we will plan sub-projects that allow us to examine the treatment outcomes of each cancer type. Our real-world evidence could complement the findings of traditional RCTs and provide valuable information for future research directions, policy initiatives, medical guidelines, and clinical decision-making. The research results will be published in peer-reviewed journals and presented at both national and international scientific conferences, as well as patient organizations.

### Strengths and Weaknesses

This nationwide cohort study will include a large number of patients from community-based general hospitals only, not including cancer centers or academic institutions. The cohort will comprise a certain number of patients who are unable to participate in clinical trials owing to their age, poor performance status, or complications. Therefore, this study could provide more realistic clinical data with a lower rate of loss to follow-up.

Moreover, our study will simultaneously use demographic data and treatment details, as well as the National Cancer Registry, DPC, prescription, and laboratory test data. In particular, few large-scale databases include performance status and laboratory data, which would be major considerations regarding prognostic factors. In addition, by using the DPC data, detailed information on emergency hospitalization, length of hospital stay, concomitant complications, medical costs, and outcomes can be obtained, allowing for a review of the quality and cost-effectiveness of medical care. Prescription data would be useful for evaluating drug–drug interactions. Relative dose intensity can also be computed. Individual analyses using institutional or national databases are very common; however, to the best of the authors’ knowledge, this is the first large-scale cohort study that can simultaneously handle comprehensive RWD.

In this study, it will be difficult to extract data without using a unified description format or sentences as descriptions in ordinary medical records. For instance, the time to treatment failure is easy to obtain, although there is no distinction between refractory and intolerant. Progression-free survival and recurrence-free survival can only be obtained through direct checks of medical records, although they are less reliable in retrospective studies [25]. In the same context, direct access to medical records is mostly required to obtain information on genetic mutations [26], which becomes an essential step when conducting standard treatment for many cancer types because the Japanese National Cancer Registry does not contain it, and most results of genetic alteration tests are included as images after outsourcing to testing companies. With regard to adverse events, laboratory data can be easily extracted and evaluated; however, it is difficult to handle imaging findings, such as intestinal lung disease, or subjective symptoms, such as nausea and vomiting. The electronic medical record system is equipped with software that evaluates and inputs according to CTCAE, but it is not sufficiently widespread during the current study period. Similarly, there is no provision for the patients’ quality of life scores to be routinely entered in the electronic medical record; thus, it is impossible to extract such data from our medical records, although its usefulness has been established [27]. These limitations indicate if there is good or poor compatibility among sub-projects for each type of cancer and treatment.

## 4. Ethics and Dissemination

All procedures performed in this study will be in compliance with the “World Medical Association Declaration of Helsinki” (revised in October 2013) and “Ethical Guidelines for Medical Research for Humans” (2014 Ministry of Education, Culture, Sports, Science and Technology, Ministry of Health, Labor and Welfare Notification, revised in February 2017).

Because this study will be carried out using existing information held by one’s own research institution, it is not always necessary to receive informed consent from the patients; however, information about the study, including the purpose of use of the information, will be notified or disclosed to the patients. Therefore, documents approved by the Joint Ethics Review Committee will be posted on the website of the Tokushukai Medical Group. We will publicly announce the research content and accept any requests for refusal to use the personal data. The patient will never be penalized if they decide to refuse participation.

When processing information associated with the conduct of the research, a number unrelated to the individual’s personal information (research subject identification code) is assigned in advance and managed as anonymization, and sufficient consideration is given to the protection of the research subject’s confidentiality. The correspondence table is strictly stored by the electronic medical record manager (Tokushukai Information System Inc.) and personal information manager. This number is used when sending information to the Department of Medical Oncology and Hematology, Kobe University Hospital, and deCult Co., Ltd. and gives due consideration to ensure that the personal information regarding the research subjects will not be disclosed outside the hospital. In addition, when the principal investigator publishes the information obtained in this study, care will be taken not to include information that can identify the research subjects.

## Figures and Tables

**Figure 1 healthcare-10-02146-f001:**
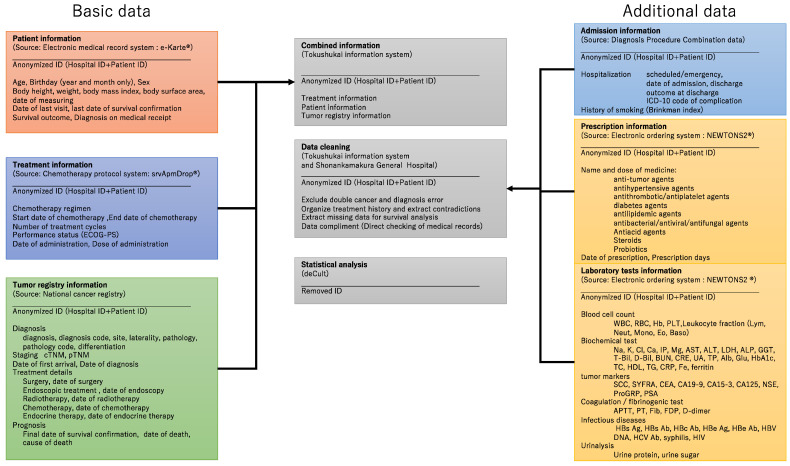
Data collection overview.

**Table 1 healthcare-10-02146-t001:** Inclusion and exclusion criteria.

Inclusion Criteria
Patients with pathologically or radiologically diagnosed cancer Age ≥20 years Receiving centrally registered systemic therapy at DPC hospitals within the Tokushukai Medical Group from 1 April 2010 to 31 March 2020
Exclusion criteria
Patients who chose to opt out Simultaneous and metachronous double cancer with disease-free interval of ≤5 years, except for carcinoma in situ/intramucosal cancer cured by local treatment or endoscopic lesions diagnosed as carcinoma in situ Inadequate data extraction (e.g., missing details on past cancer treatment at other hospitals or individual patient data)

**Table 2 healthcare-10-02146-t002:** Laboratory test information.

Blood cell counts	White blood cell (WBC), red blood cell (RBC), hemoglobin (Hb), platelet (PLT), leukocyte fraction, lymphocyte (Lym), neutrophil (Neu), monocyte (Mono), eosinophil (Eo), and basophil (Baso)
Biochemical tests	Sodium (Na), potassium (K), chloride (Cl), calcium (Ca), inorganic phosphorus (IP), magnesium (Mg), aspartate aminotransferase (AST), alanine aminotransferase (ALT), lactate dehydrogenase (LDH), alkaline phosphatase (ALP), γ-glutamyl transpeptidase (GGT), total bilirubin (T-Bil), direct bilirubin (D-Bil), blood urea nitrogen (BUN), creatinine (CRE), uric acid (UA), total protein (TP), albumin (Alb), blood glucose (Glu), hemoglobin A1c (HbA1c), total cholesterol (TC), HDL cholesterol, triglyceride (TG), C-reactive protein (CRP), iron (Fe), and ferritin
Tumor markers	Squamous cell carcinoma-related antigen (SCC), cytokeratin 19 fragment (SYFRA), carcinoembryonic antigen (CEA), carbohydrate antigen 19-9 (CA19-9), carbohydrate antigen 15-3 (CA15-3), carbohydrate antigen 125 (CA125), neuron-specific enolase (NSE), pro-gastrin-releasing peptide (ProGRP), and prostate specific antigen (PSA)
Coagulation/fibrinogenic tests	Activated partial thromboplastin time (APTT), prothrombin time (PT), fibrinogen (Fib), fibrin/fibrinogen degradation product (FDP), and D-dimer
Infectious disease markers	Hepatitis B surface (HBs) antigen, HBs antibody, HB core antibody, HBe antigen, HBe antibody, HB virus DNA quantification, HC virus antibody, syphilis, and HIV
Urinalysis parameters	Urine protein qualitative, urine sugar qualitative

**Table 3 healthcare-10-02146-t003:** Sample size of each cancer type.

Biliary tract cancer	800
Breast cancer	10,000
Colorectal cancer	10,000
Esophageal cancer	1000
Gastric cancer	5000
Gastrointestinal stromal tumor	500
Kidney cancer	350
Liver cancer	500
Lung cancer	10,000
Ovarian cancer	1500
Prostate cancer	5000
Pancreatic cancer	2500
Uterine body cancer	350
Uterine cervix cancer	350
Urothelial cancer	1000

## Data Availability

Not applicable.

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
