# Peer review of "Real-World Outcomes of Systemic Therapy in Japanese Patients with Cancer (Tokushukai REAl-World Data Project: TREAD): Study Protocol for a Nationwide Cohort Study"

_healthcare, 2022, doi:10.3390/healthcare10112146_

Round 1

Reviewer 1 Report

Quite original study idea, too why there are not many similar studies; not me they liked the abstract very much, since the scoop of the study is not very clear, so the concept should be strengthened. The second problem concerns the bibliography, too small, albeit updated. For the rest I liked the article.

Author Response

We appreciate your effort and valuable comments. According to the reviewer’s suggestion, we have clarified the purpose of this study, and we have almost doubled the number of relevant references (14 to 27).

  • Previous

To evaluate whether the results of clinical trials are carried over to the real-world, we conducted an exploratory cohort study using RWD to investigate temporal trends in treatment patterns for advanced cancer as well as treatment outcomes and prognostic factors that influence OS.

Here, we report the protocol of the study that will generate data addressing several research questions through several sub-projects for each type of cancer and treatment.

  • Revised (Lines 80-84)

Here, we report the main protocol of the study that will generate data through several sub-projects for each type of cancer to assess whether the results of clinical trials extend to the real world. Moreover, the data will enable examination of temporal trends in treatment patterns, as well as treatment outcomes and prognostic factors that affect overall survival.

Reviewer 2 Report

The study protocol for this nationwide cohort study presents the methodology collect RWD on systemic therapy for Japanese patients with cancer in the Tokushukai Medical Group with this merit of scale.

For this study, the primary endpoint is overall survival, whereas the secondary end-points are the time to treatment failure, recurrence-free survival, and adverse events, evaluated based on the JCOG version. Can you extract also information about the quality of life of the patients before and after the treatment?

Figure 1 must be sharper.

Author Response

We appreciate your effort and thank you for pointing out this important issue. Unfortunately, it is difficult to extract information on quality of life in this project, although its usefulness has been demonstrated. We have clarified this point as a limitation in the ‘Strengths and Weaknesses’ section. Further, we have replaced Figure 1 with the one with higher resolution.

  • Previous

(none)

  • Revised (Lines 259-261)

Similarly, there is no provision for the patients’ quality of life scores to be routinely entered in the electronic medical record; thus, it is impossible to extract such data from our medical records, although its usefulness has been established [27].

Reviewer 3 Report

The work is devoted to large-scale and long-term research, which undoubtedly is its value for the scientific community. However, this work seems unfinished. There is no clear formulation of the goal of this article, as there are no conclusions regarding its achievement. The work actually only reports on the conducted research, but what results and what conclusions are valuable for science and practice are not presented by the authors.The "Discussion" section does not contain a discussion and comparison of the obtained results. Figure 1 is low resolution.

Author Response

We appreciate your effort and constructive comments. We would like to confirm that we have presented a study protocol in this manuscript; hence, we have not obtained any results yet. We plan to conduct several sub-projects to evaluate treatment outcomes of each cancer type. Upon obtaining results, we will discuss them while comparing with relevant previous data. To clarify this point and expected impacts, we have added a sentence in the discussion section and changed the last sentence of the abstract. In addition, we have now provided Figure 1 in a higher resolution.

  • Previous

The results of the study will be published in peer-reviewed journals and presented at both national and international scientific conferences as well as patient organizations.

  • Revised (Lines 31-32)

Our findings provide important insights for future research directions, policy initiatives, medical guidelines, and clinical decision-making.

  • Previous

Our real-world evidence could complement the findings of traditional RCTs.

  • Revised (Lines 218-222)

Following the main protocol, we will plan sub-projects allowing us to examine the treatment outcomes of each cancer type. Our real-world evidence could complement the findings of traditional RCTs, and provide valuable information for future research directions, policy initiatives, medical guidelines, and clinical decision-making.

Round 2

Reviewer 3 Report

The article has become better after eliminating the shortcomings. I recommend publishing this work.